# The COVID-19 Pandemic Situation in Malaysia: Lessons Learned from the Perspective of Population Density

**DOI:** 10.3390/ijerph18126566

**Published:** 2021-06-18

**Authors:** Siew Bee Aw, Bor Tsong Teh, Gabriel Hoh Teck Ling, Pau Chung Leng, Weng Howe Chan, Mohd Hamdan Ahmad

**Affiliations:** 1Faculty of Built Environment and Surveying, Universiti Teknologi Malaysia, Skudai 81300, Johor, Malaysia; awsiewbee@gmail.com (S.B.A.); pcleng2@utm.my (P.C.L.); b-hamdan@utm.my (M.H.A.); 2School of Computing, Faculty of Engineering, Universiti Teknologi Malaysia, Skudai 81300, Johor, Malaysia; cwenghowe@utm.my

**Keywords:** population density, coronavirus, COVID-19, Pearson correlations, spatio-temporal, non-pharmaceutical measures, Malaysia

## Abstract

This paper attempts to ascertain the impacts of population density on the spread and severity of COVID-19 in Malaysia. Besides describing the spatio-temporal contagion risk of the virus, ultimately, it seeks to test the hypothesis that higher population density results in exacerbated COVID-19 virulence in the community. The population density of 143 districts in Malaysia, as per data from Malaysia’s 2010 population census, was plotted against cumulative COVID-19 cases and infection rates of COVID-19 cases, which were obtained from Malaysia’s Ministry of Health official website. The data of these three variables were collected between 19 January 2020 and 31 December 2020. Based on the observations, districts that have high population densities and are highly inter-connected with neighbouring districts, whether geographically, socio-economically, or infrastructurally, tend to experience spikes in COVID-19 cases within weeks of each other. Using a parametric approach of the Pearson correlation, population density was found to have a moderately strong relationship to cumulative COVID-19 cases (*p*-value of 0.000 and R^2^ of 0.415) and a weak relationship to COVID-19 infection rates (*p*-value of 0.005 and R^2^ of 0.047). Consequently, we provide several non-pharmaceutical lessons, including urban planning strategies, as passive containment measures that may better support disease interventions against future contagious diseases.

## 1. Introduction

The highly infectious novel coronavirus disease, more commonly known as COVID-19, is an ongoing pandemic that has strained health facilities worldwide. As of 31 March 2021, it has infected 129 million and killed 2.83 million people; up to that date, Malaysia has recorded 345,500 infections and 1272 deaths since the first case reported on 24 January 2020.

The inventions of COVID-19 detection kits, such as RTK-Ag and RT-PCR, as well as vaccines, such as Pfizer-BioNTech, AstraZeneca, or Sinovac, while timely, are overshadowed by the continuous mutation of the SARS-CoV-2 virus. The UK strain B.1.1.7, South African strain B.1.351, and Brazilian strain P.1 were reported to be more infectious [1]. More recently, new strains reported in France and Finland were reported to be undetectable by standard RTK-PCR tests [2]. Existing vaccines only reduce the risk and severity of the infection [3].

Furthermore, vaccines and test kits are reactive interventions that tackle COVID-19 specifically, and does not prepare the world for the next pandemic. SARS-CoV-2 is the latest highly pathogenic human coronavirus over the past two decades, after SARS-CoV in 2002 and MERS-CoV in 2012 [4]. These coronaviruses cause respiratory diseases, like influenza viruses, but they are much more contagious and dangerous [5]. The viruses are transmitted in several ways [5], including direct human contact, air-borne droplets [6], and fomites, which refer to surfaces that can carry these viruses for varying periods of time, from hours to days [7,8]. SARS-CoV-2, for example, can live on metal, glass, and ceramics for five days; wood for four days; and on stainless steel and plastics for two to three days [9]. The risk of rapid human-to-human transfer of the virus through respiratory droplets and close contact is high [10].

To combat COVID-19, governments worldwide mandated the use of face masks and hand sanitisers, enforced social distancing, and then enacted lockdowns that restricted the movement of citizens save for essential workers. The combination of these measures abated the spread of COVID-19, most notably when full lockdowns were in effect, but at a heavy economic cost: in Malaysia, the first Movement Control Order between March and May 2020 was estimated to have cost Malaysia approximately RM2.4 billion, while the second round between January and March 2021 may incur more than RM200 million in losses [11]. Premature relaxation of movement-restriction measures, and the resultant rise in mobility rates, translated into spikes of COVID-19 transmissions that had to be promptly addressed.

The adverse socio-economic impacts of lockdowns make them unfeasible as long-term mitigation measures for COVID-19 and future similar outbreaks. The World Health Organisation recommends people to avoid close-contact settings and crowded and confined or closed spaces. This is theoretically more challenging to achieve in dense urban areas than sparse rural areas, as incidents of face-to-face interactions potentially increase [12]. In other words, population density or crowdedness taking into account both population size and spatial consideration serves as a crucial factor in determining the severity or transmission risks of COVID-19 and it will be the focal discussion in this study. Prior to explaining whether and how population density influences COVID-19 severity in terms of the number of infections and fatalities, based on the review of both past and recent literature covering COVID-19 and other contagions, since Malaysia has been selected as a case study, to provide a general background understanding, its COVID-19 situation is described first in the next section.

## 2. COVID-19 Status in Malaysia

Malaysia has undergone three major waves of COVID-19 outbreaks since the announcement of the pandemic in March 2020. The first wave, spanning between 25 January to 16 February 2020, precluded the official World Health Organisation (WHO) classification of COVID-19 as a pandemic; the second wave lasted between February 27 and June 30, 2020; and the third wave has been ongoing since 8 September 2020 [10].

The first Movement Control Order (MCO) and its subsequent phases, the Conditional Movement Control Order (CMCO) and Recovery Movement Control Order (RMCO), succeeded in bringing daily COVID-19 cases down to low double- or single-digit reports by June 2020 [13]. This situation prevailed until early September 2020, when cases surged due to the emergence of two major clusters and the more infectious D614G-type mutation of the SARS-CoV-2 virus [10]. Several possible factors might have exacerbated the situation, including increased human mobility caused by politics, increased COVID-19 testing rates, and waning concern about the pandemic among Malaysians [10]. The third wave has been by far the most severe of the three waves, having consistently claimed more than 1000 new cases daily since 3 November 2020 [13]. Fortunately, the COVID-19 mortality rate in Malaysia is comparatively lower than many other countries, with only 1272 deaths recorded out of 345,000 confirmed cases as of 31 March 2021, or 0.37% of detected cases [14]. The following are the figures of all three waves in terms of cumulative cases and mortality cases. According to the official statistical data from the Ministry of Health, Malaysia [14], the first wave (25 January 2020–16 February 2020) of the COVID-19 pandemic in Malaysia recorded a total of 22 infected individuals and 0 deaths. While for the second wave (26 February 2020–30 June 2020), cumulative cases of 8616 infected individuals and 121 deaths were recorded. Lastly, the number of the third wave (September 8–till present) has increased exponentially, where it has recorded a total of 103,551 infected individuals and 343 deaths (as of 31 December 2020).

This study analyses, at the district level, the cumulative confirmed cases and rate of cumulative confirmed COVID-19 cases in Malaysia. The country comprises 13 states, including three federal territories, spread across East Malaysia (Sabah and Sarawak) and West Malaysia (Peninsular Malaysia). Together, there are 143 local districts in the country (refer to Figure 1).

The Malaysian government started publishing reports of COVID-19 cases and mortalities from the day the COVID-19 pandemic was announced. Both state- and federal-level data were reported, although there were differences in the methods of delivery. At the initial stages of the pandemic, all states reported daily cases, but as the pandemic progressed, several states switched to a 14-day average detailed report for every district, while the Ministry of Health Malaysia continued to report daily state-level statistics. As the 14-day moving average method was only fully adopted after 7 May 2020, district-level tabulation started with baseline cumulative figures recorded between 25 January and 7 May 2020, and every 14 days thereafter. The 7 May 2020 coincided with the first Conditional Movement Control Order (CMCO); from 4 May 2020, Malaysians were allowed to travel for work in addition to essential trips for food and necessities. The number of positive cases plateaued with occasional minor spikes until mid-September or early October 2020, coinciding with agitated human mobility caused by a relaxation of travel restrictions and a political election in the state of Sabah.

## 3. COVID-19 and Population Density

The impact of population density on infectious diseases and their transmission rates is neither immediately apparent nor uniform worldwide. In theory, density brings about higher changes of close contact and interactions between inhabitants [12,15]. Some studies reported positive correlations [15,16,17], while others reported flat or negligible correlations [12,18]. World Bank data indicated an absence of significant causality between density and COVID-19 transmission [19]. Past studies found a stronger correlation between connectivity and COVID-19 than density due to well-developed economic, social, and transport linkages within and beyond urban centres [12]. Urban centres also tend to have better-developed facilities and amenities per capita with easier access to healthcare. Further studies indicate that dense locations may result in earlier COVID-19 transmission outbreaks but not necessarily with increased cases or deaths [20].

Population density, taken as the total population of a district or state over its total area, can be a more accurate indicator of urban density than population distribution. The Department of Statistics Malaysia [21] indicates that national population density was 86 persons per square kilometre in 2010—the year of the latest population census—with large discrepancies between the most and least densely populated states. Selangor was only the fifth densest state despite being the most populous [21]. The population density of the Federal Territory of Kuala Lumpur, the densest state, is almost 340 times higher than that of Sarawak, the least dense state [21].

Some studies argue that this definition of population density is too simplistic; the number of people in an area fluctuates throughout the day, depending on the purpose and variety of the neighbourhood [22]. Factories and offices can have higher population densities during operating hours, although they disperse into residential areas at the end of the workday, a phenomenon known as crowding. The recent third wave of COVID-19 infections in Malaysia has been attributed to temporary crowding in such workplaces [23,24].

The COVID-19 outbreak aboard the Diamond Princess cruise ship, lasting between 21 January and 19 February 2020, is an intriguing case study of COVID-19 transmission in a restricted area with a high number of common spaces that encourage crowding. The Diamond Princess hastened to dock and quarantine at a port in Japan upon the identification of an index case between 21 January and 25 January 2020. Independent of further infections from external sources, passengers on the ship faced quarantine in their cabins with limited, but daily, access to common facilities with limited sanitisation measures in place. By 20 February 2020, when all 3700 of the passengers and crew were evacuated, 17% of the occupants of the Diamond Princess had tested positive for COVID-19 [25].

In the United States, some studies have determined that while places with high population density tend to experience earlier and higher numbers of COVID-19 outbreaks, this positive relationship declines over time [16,20]. Many countries, including Malaysia, first detected the onset of positive COVID-19 cases in densely populated cities, which had an increased tendency to experience higher rates of human mobility. Based on data obtained from 14-day moving charts, the occurrence of new COVID-19 cases within and beyond urban epicentres is analysed to determine whether a similar proliferation pattern is observed in Malaysia.

Population density can have a positive correlation with the number of high-rises in an area, given that high-rise buildings, primarily for residential and commercial purposes, provide technical and economic benefits for an area that is facing high demand and land scarcity [22]. Malaysia is trending in that direction, with compact high-rise developments making up the bulk of recent stratified developments in the country [26]. These high-rises are characterised by a relatively high percentage of common spaces, such as corridors, lifts, staircases, and lobbies. The physical proximity induced by such developments has been instrumental in the transmission of previous pandemics, such as the 1918 Spanish influenza.

However, it must be noted that other factors besides density also play a role in the mitigation of COVID-19 transmission. Highly dense countries, such as Singapore and Hong Kong, demonstrated lower confirmed COVID-19 mortality rates than many less-dense countries like Russia and Germany. Swift and strict government-level interventions, a high level of obedience by citizens, a robust and accessible healthcare system, and aggressive testing and quarantining measures are among the factors credited for the lower transmission rates in those countries. Based on the above synthesis, we found that there are inconsistent findings or lack of consensus about the effects of population density on the COVID-19 transmission; questions of whether or how significant population density impacts the severity of COVID-19 remain unanswered, particularly for the case of Malaysia as there no empirical research has been undertaken so far. Therefore, it is necessary to ascertain the weightage or significance of population density as a factor in contagious disease transmission in order to improve passive urban resilience against future highly infectious pandemics.

## 4. Methods

### 4.1. Data Collection

The dataset in this study combined data on COVID-19 cases, population density, the probability of crowding through compact development lifestyles, and spatio-temporal spread of COVID-19 transmissions at a district level in Malaysia. The study investigated these factors between 25 January 2020 and 31 December 2020, comprising 342 days of COVID-19 data. The data were extracted from the Ministry of Health Malaysia’s (MOH) website (http://covid-19.moh.gov.my, accessed on 28 March 2021) [14]. The delineation of Malaysian districts is taken in accordance with local authority jurisdictions. In the event of data discrepancy between federal- and state-level data, federal-level data were used.

### 4.2. COVID-19 Cases

Primary data on COVID-19 cases and deaths by district in Malaysia was extracted from reports published by MOH on their official website (http://covid-19.moh.gov.my, accessed on 28 March 2021) [14]. As the cumulative COVID-19 figures are likely to be distorted by local testing capacities, delays in reporting, unreported cases, and undetected asymptomatic carriers of the virus, accurate recording of infection rates within a community is challenging. The COVID-19 mortality rate, a measure considered more reliable than daily cases [20,27], is discounted due to the statistically insignificant mortality rate in Malaysia, which is approximately 0.37% as of 31 March 2021.

The data reported by MOH has evolved several times since the start of the pandemic, and the report format for each state is non-uniform. Out of the 13 states and 3 federal territories in Malaysia, Penang, Perak, Negeri Sembilan, Kelantan, and Sarawak have adopted a 14-day moving average for COVID-19 reports, while the others still report daily and cumulative cases at varying degrees of detail (state-wide or district-wide). The 14-day moving average was utilised for the data analysis for these five states in the absence of complete MOH data.

### 4.3. Population Density

The population of each Malaysian district was obtained from the latest population census for 2010. This is in view of the delayed 2020 Population and Housing Census caused by the onset of the COVID-19 pandemic. The figures were then divided with the geographical size of the district to derive the population density. Based on population density data from the Department of Statistics Malaysia (DOSM), Penang represents the denser state, while Perak, Negeri Sembilan, Kelantan, and Sarawak represent the less-dense states.

### 4.4. Data Analysis

The data were analysed descriptively based on observation and inferentially using SPSS in order to determine the correlation between population density and the number and rate of positive COVID-19 cases. The cumulative confirmed COVID-19 cases and confirmed COVID-19 infection rates were plotted against the population density of every district in Malaysia. The data were interpreted using Pearson’s coefficient. As a rule of thumb, we considered the results to have a high degree of correlation if the R-value lay between ±0.50 and ±1; a moderate degree if it lay between ±0.30 and ±0.49; and a low degree if it lay below ±0.29. We further refined the analysis via the coefficient of determination in the regression models, expressed as R-squared (R^2^) values, to study the variance in the COVID-19 cases. From a range between 0% and 100%, a higher R^2^ value denotes a higher explanation of the variability in the response data around its mean. By reading R-values in conjunction with R^2^ values, we obtained the degree of changes that could be explained by the compared data. In addition to this, the temporal diffusion of confirmed COVID-19 cases from the initial COVID-19 epicentres was charted to determine the rate and spatial reach of the outbreak over time.

## 5. Results and Findings

### 5.1. Spatio-Temporal Spread of COVID-19

An analysis of the fluctuations in confirmed COVID-19 cases in every district was carried out to ascertain the virulence and decay of the virus across distances. From September 2020, the number of cases across the country increased exponentially in many districts. The most populous districts in each state were among the worst hit at the beginning. Among many states, mainly three states and their districts were reported as instances covering both the west and east region of the country.

Penang experienced a major surge in a fortnight, between 9 October and 22 October 2020, where its most populous Timur Laut district recorded 402 new cases, while the less populous Barat Daya and Seberang Perai Tengah districts recorded only one new case, as shown in Table 1. Over the weeks that followed, the growth in the confirmed number of cases was tamer on mainland Penang (comprising Seberang Perai Utara, Seberang Perai Tengah, and Seberang Perai Selatan districts) while the numbers increased almost constantly every fortnight in the two districts (the Timur Laut and Barat Daya districts). The number of new cases in Seberang Perai Utara was steady and had actually decreased by 31 December 2020. In comparison, the Barat Daya district saw its new cases soar in the weeks following the record high of 402 cases detected in the neighbouring Timur Laut district. By end-November 2020, it had overtaken the Timur Laut district in the number of new cases reported, and the trend continued until the end of the year. The Timur Laut district of Penang has a high concentration of jobs in various sectors, including a large industrial zone straddled between the Timur Laut and Barat Daya districts. Economic forces encouraged daily human mobility from the Barat Daya district into the Timur Laut district, where people were at a higher risk of cross-infection of the virus.

A slightly different pattern was observed in the state of Perak, as shown in Table 2, in which the initial resurgence of COVID-19 occurred in the Larut and Matang district, where Perak’s second-largest city, Taiping, is located. Over the following weeks, the number of cases in several nearby districts, such as Kuala Kangsar and Kinta, increased, and continued increasing even after the number of cases in Larut and Matang declined. The pattern is almost similar across all states: a more populous district would experience a resurgence, and in the following weeks, several nearby districts would follow suit, although the rates of COVID-19 reports may fluctuate.

Sarawak and Perlis are exceptions. While Sarawak’s most populous district of Kuching experienced more vigorous COVID-19 activity throughout the study period, most new cases were contained to Kuching, with only occasional low reports of confirmed COVID-19 reports in other districts, as shown in Table 3, Table 4 and Table 5. There was a short burst of cases in Bintulu and Miri, the second and third most populous districts in Sarawak, but they were soon quelled. Other districts reported singular cases sporadically. Like Sabah, the rough and poorly connected terrain in Sarawak may have contributed to reduced human mobility between districts and therefore a containment of COVID-19 virulence within hotspots. The secluded location of Perlis on the northernmost tip of Peninsular Malaysia, as well as its low population density, may have contributed to its low number of cases throughout the pandemic.

### 5.2. Population Density and COVID-19 Cases in Malaysia

The study period encompasses two COVID-19 outbreak waves as well as a full cycle of MCO, CMCO, and RMCO in Malaysia. By the end of the study period, COVID-19 cases had peaked twice. The first peak occurred in April 2020 while the second is still ongoing by 31 December 2020. It reached a record number of 4284 daily cases on 3 February 2021, a timeframe that is beyond the scope of this study, but tapered thereafter under the effects of a second MCO enforced on 11 January 2021.

Overall, Figure 2 shows a strong positive correlation between the population density of a district and the cumulative confirmed cases of COVID-19. We determined an R-value of 0.644, which indicates a moderately strong correlation between the two variables (i.e., p-value < 0.005). Generally, the district of Kuala Lumpur, a federal territory and the capital city of Malaysia, reported both the highest population density per square kilometre and the highest number of confirmed cumulative cases up to 31 December 2020. It reported almost three times the total number of confirmed cases of the district of Petaling in neighbouring Selangor, which is approximately half as dense as Kuala Lumpur.

Not all states conform to this pattern. The Timur Laut district of Penang state, located in the northern region of Peninsula Malaysia, is the second most dense district in the country, yet it recorded approximately 80% less cases than Kuala Lumpur within the same timeframe. Districts in closer geographical proximity to Kuala Lumpur, such as Petaling, Klang, and Hulu Langat, reported relatively higher cumulative confirmed cases of COVID-19 despite being several magnitudes less dense than the capital city. The state of Sabah also appears to be an anomaly with its low density and high prolonged COVID-19 incidence rates, although its situation may have been exacerbated by crowding and travel-related transmissions, compounded by poorer healthcare infrastructure.

These patterns, by themselves, do not form conclusive evidence that urban density causes higher COVID-19 infections. As COVID-19 can go undetected in asymptomatic carriers, or sometimes go undocumented due to the mildness or limited number of symptoms, including ignorance in the early stages of the pandemic, Malaysia’s largely symptom-based screening approach may be insufficient to capture the true situation at hand. However, the result returned an R^2^ value of 0.415, revealing that population density is able to explain over 40% of the cumulative COVID-19 cases in Malaysia. The correlation results show that population density is a key factor in estimating the cumulative confirmed cases of the disease.

We further refined the result by plotting the transmission rate of confirmed COVID-19 cases, taken as the cumulative number of confirmed COVID-19 cases up to 31 December 2020, against the total 2010 population density of each district. By analysing the infection rates, we were able to observe the influence of other underlying factors, such as socio-economic backgrounds, that may have affected the resultant cumulative COVID-19 cases in a district.

The graph returned an R-value of 0.216 (Figure 3). This implies that even upon refinement of the data, which has been associated with other factors not explicitly included within the scope of this study, the relationship between population density and COVID-19 infection rates is still moderate. Unlike with cumulative confirmed COVID-19 cases, population density could only explain approximately 5% of the COVID-19 infection rates, with an R^2^ value of 0.047. The influence of other underlying factors may have lowered the strength of population density as a variable of COVID-19 infection rates.

When the number of infections is taken as a percentage of the total population, a large population does not necessarily correlate with a higher rate of COVID-19 infections. Rather, several districts in Sabah have the highest rates of COVID-19 cases, despite Sabah being among the least populous states in Malaysia. On the other hand, urban districts, such as Kuala Lumpur, Timur Laut district of Penang, and Petaling in Selangor, recorded lower than average rates. The Kota Kinabalu district in Sabah recorded nearly double the rate of confirmed cases in comparison to the Klang district in Selangor, despite having almost similar population densities.

There may be several explanations for this scenario. Unlike Kuala Lumpur, Penang, or Selangor, which are established urban centres with well-developed infrastructure, facilities, and amenities, Sabah is still developing. Urban centres tend to be equipped with a larger number of healthcare facilities that may be better equipped to handle COVID-19 cases. Any deficit may be overcome by accessing facilities in a neighbouring district via a thorough transportation network. This may mitigate the heightened risk of COVID-19 contagiousness found in such populous areas. As of 31 December 2020, Kuala Lumpur has 162 COVID-19 testing facilities in the form of private hospitals, private clinics, and private ambulatories, compared to only 62 such facilities in Sabah, according to data sourced from the MOH website. This means Kuala Lumpur has one COVID-19 testing facility per 2.51 square kilometres, in comparison to Sabah’s one facility per 1187.42 square kilometres. Kuala Lumpur has one facility per 9806 people to Sabah’s one facility per 1187 people. However, while Kuala Lumpur has 162 facilities to serve its single district, Sabah’s three worst hit districts, namely Kota Kinabalu, Putatan, and Penampang, have a total of 21 facilities to serve them, with a large majority centred in Kota Kinabalu.

Part of this disparity can be attributed to the geographical terrain in Sabah, which is largely undeveloped. The distribution of communities in the state can be sparse with poor transportation networks, thus impeding COVID-19 detection and treatment of suspected cases. By contrast, Kuala Lumpur is one of the most developed states in the country. The concentration of COVID-19 testing facilities in the state makes it easier to detect and treat COVID-19, albeit harder to contain due to high population density. The high infection rates in Sabah despite its limited ability to detect COVID-19 makes the situation in the state even more alarming. The combination of these factors caused Sabah to be the worst-afflicted state until 12 January 2021, after which it was overtaken by Selangor [28].

The correlations between population density, cumulative COVID-19 cases, and COVID-19 infection rates are summarised in Table 6.

## 6. Discussion: Lessons Learned and Recommendations

Symptomatic COVID-19 patients are more contagious regardless of the presence of close contacts [29], but asymptomatic patients run higher risks of spreading the virus to the general populace as they move around [30]. Asymptomatic infections are estimated to be 20% [31]. Even so, asymptomatic patients were the sources of some of the initial clusters at the early stages of the pandemic in Malaysia [32]. In both scenarios, patients with co-morbidities tend to experience more severe symptoms of COVID-19 [33,34].

In the absence of widespread vaccination against COVID-19, which is currently underway in Malaysia, mask-wearing and social-distancing are probably among the most effective ways to limit the spread of the SARS-CoV-2 virus. As countries progress, urbanise, and encourage migration to cities, the threat of the next contagious disease is always a point of concern. The virulence of the COVID-19 pandemic is devastating but certainly not unprecedented; almost a century ago, the 1918 Spanish flu also ravaged populous areas and killed hundreds of thousands before it receded.

The fast proliferation of highly infectious diseases highlights the urgent need for passive containment strategies that may help limit the spread of contagious viruses without resorting to drastic lockdowns. Malaysia is fortunate to have a low COVID-19 mortality rate despite undergoing several outbreaks thus far. Among these fatalities, a majority of them had co-morbidities that may have exacerbated COVID-19 symptoms.

The findings of this study agree with those by Carozzi et al. [20] for the US: dense locations experience earlier and more serious COVID-19 outbreaks. Dense areas that are also well-connected economically, socially, and logistically risk spreading the virus to neighbouring districts. Good networks allow districts to share resources to handle outbreaks. When there is poor healthcare infrastructure across a large geographical area, a situation that may limit the ability of a district to detect and isolate COVID-19 victims, this can result in prolonged outbreaks as undetected victims continue spreading the virus in the community.

### 6.1. Self-Contained Urban Planning

More recent findings have revealed workplaces and institutional facilities to be the sources of new COVID-19 clusters. Compact living and working is inevitable in urban centres due to land scarcity and high demand. As such, one possible method to contain contagious diseases, from an urban planning standpoint, may be to ensure that citizens have ample options within their chosen district to live, work, and play. The generation of economic opportunities in relative proximity to residential areas can encourage citizens to live near where they work. By condensing such leisurely and commercial activities across a certain distance or district boundary, essential travel across districts may be reduced. This increases the chances of confining a contagious disease within a small area, contingent on swift early detection and containment of the disease.

Current Malaysian spatial policy adopts a macro-first view: the broad national physical plan gets distilled into structural plans, then into local plans, both of which describe land-use opportunities in the form of zoning. Restrictions on land use are also placed on individual lots, as stated on land titles. However, developers are given the opportunity to rezone and change the land use of the land. Developers with strong financial standing are then able to purchase large swathes of land, form land banks, and then convert them into the desired zoning and land use category under the guise of township master-planning, thus overriding the local plan, especially if the local plan has yet to be gazetted. It gives developers the power to make decisions based on perceived economic viability rather than ease of access by bicycle or on foot. The scale of such masterplans is measured by the travel distance of a car, resulting in disproportionate provisions of residential and commercial components, as well as multi-hectare sections of residential-only or commercial-only areas.

In urban areas, zoning restrictions have given rise to mixed developments that combine residential, commercial, and recreational components on the same site. Although proposing mixed developments is one answer to condensing living and working environments, the variety of job opportunities generated in such developments is limited. The commercial components in many mixed developments tend to be geared towards retail and leisure, while the rental or purchase rates of such commercial units make them less appealing for businesses that do not rely on foot traffic, such as consultancies. Residents that do not match this economic demographic still end up looking elsewhere for job opportunities.

While urban design may have limited ability to dictate the types and varieties of businesses in an area, a more equitable ratio of workplaces to residential within a certain land area may be desirable so that people have less need for frequent long-distance essential travel.

### 6.2. Managing Crowding Risk in Dense Areas

A dense population is commonly associated with crowding. Besides considering the concentration of living and working zones, the dispersion of transient populations emerges as a key factor in reducing crowding risk, which in turn may reduce the rate of contagion through surface contact or air-borne particles. High-rises and the wide surface area of common spaces associated with them theoretically create more areas for the incubation of viruses.

While occupants of residential buildings are, by virtue of their shared location, from the same district, high-rise commercial towers encourage a large volume of people from many different areas to congregate in a single building. The same scenario can be found in labour-intensive industrial buildings. Although the crowding is temporary and limited to working hours, extended periods of exposure increase the risk of infection, as evidenced by the growing number of COVID-19 clusters at workplaces. In conjunction with master-planning closer living and working environments, further analysis on maximum crowding at workplaces merits further study.

## 7. Conclusions

Overall, this study found a moderately strong correlation between population density and higher incidences of confirmed COVID-19 cases in Malaysia. Districts that have denser populations had higher COVID-19 infection rates and higher cumulative COVID-19 cases up to 31 December 2020. Having thus identified the correlation, we may begin to devise passive strategies that may mitigate the effects of similar air- and surface-borne diseases. However, this study did not take into account the impact of sociological or governmental interventions on the management of the virus, and it must be noted that the data were collected when various movement control orders were in effect, a fact that may have affected the findings differently than a non-pandemic situation. Further studies would be required to refine the model by determining the effects of a different age, economical, and educational level of compositions of each population on COVID-19 transmission rates. Extrinsic factors, such as predominant building typologies, frequency of shared routes that facilitate high flows of people and contact points, and availability of healthcare facilities, also merit investigation.

In general, urban areas concentrate a large population over a relatively small area within which they interact, live, and work. In Malaysia and other countries, urban districts were the sources of initial COVID-19 outbreaks. The spread of the disease to neighbouring districts is wider in areas that are intensely inter-connected economically and infrastructurally, as opposed to isolated places with poor connections. However, as humans are social creatures, isolation is costly and unwieldy as a permanent measure. The severity of an outbreak is also dependent on factors, such as swift containment measures, the demographics of the population, access to healthcare, and adherence to precautionary measures (see [35,36]). As a passive strategy, the sustainability of existing city- and urban-planning policies may warrant revisiting in order to make cities more resilient against future contagions. This scope of this study was limited to the effect of population density figures on the incidence of positive COVID-19 cases. The non-uniformity of COVID-19 report formats between Malaysian states as reported on the MOH website, as well as the exclusion of imported cases, whether from overseas or from cross-state or cross-district travel, contributes to limitations in data comparisons. Regardless, the findings of this study suggest that a minimised volume of necessary trans-district travel, as well as reduced crowding risk at workplaces, may slow down the transmission of similar pandemic viruses within and beyond a pandemic’s epicentre.

## Figures and Tables

**Figure 1 ijerph-18-06566-f001:**
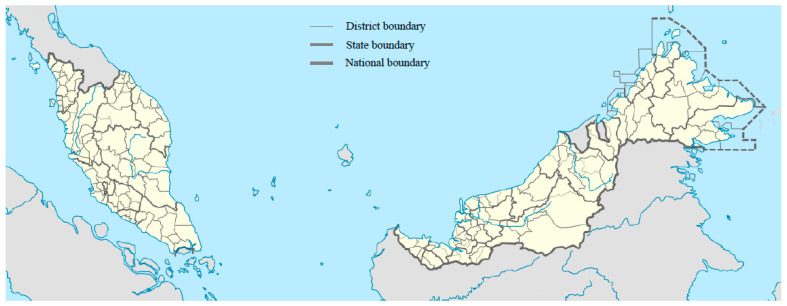
The illustration describes the spatial scale of the entire nation, state, and district in Malaysia. Malaysia is a country comprising 13 states (including three federal territories). In terms of a local district level, a total of 143 districts can be found in Malaysia. Source: Adapted from original “Malaysia Location Map with District” by Hellerick, used under CC BY 4.0.

**Figure 2 ijerph-18-06566-f002:**
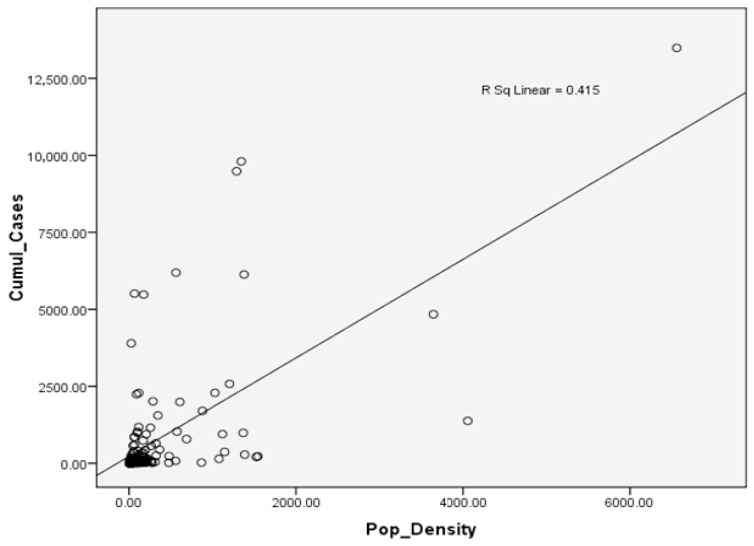
Analysis revealed a significant relationship between population density and cumulative COVID-19 cases in 2020. The model revealed an R-value of 0.644 and an R^2^ value of 0.415.

**Figure 3 ijerph-18-06566-f003:**
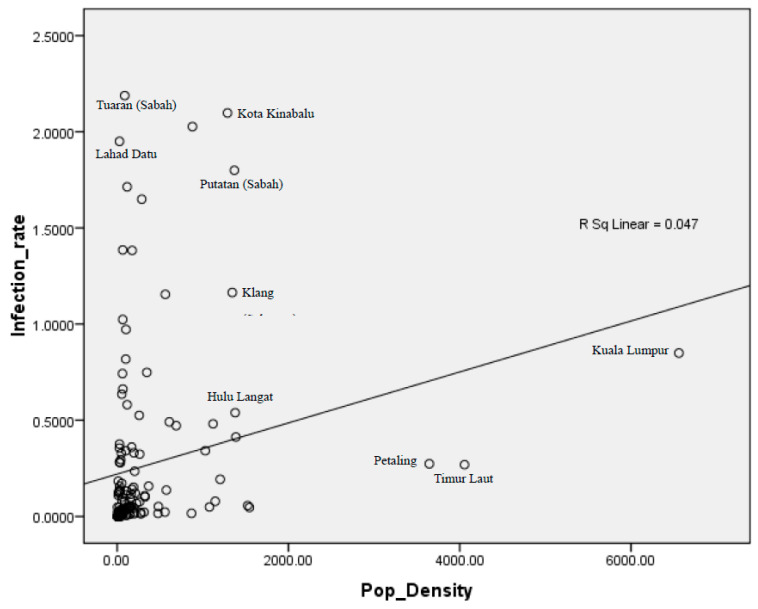
Analysis revealed a moderately strong relationship between population density and COVID-19 infection rates in 2020. The model revealed an R-value of 0.216 and an R^2^ value of 0.047.

**Table 1 ijerph-18-06566-t001:** COVID-19 cases from 25 January–21 December 2020 for five districts in the Penang State, Malaysia.

14-Days Period	Districts in Penang				
	Seberang Perai Tengah	Seberang Perai Utara	Seberang Perai Selatan	Timur Laut	Barat Daya
	Year 2010 Population Density (Person per sq km): 1524	Year 2010 Population Density (Person per sq km): 1077	Year 2010 Population Density (Person per sq km): 689	Year 2010 Population Density (Person per sq km): 4056	Year 2010 Population Density (person per sq Km): 1120
18 December–31 December 2020	42	19	65	245	252
4 December–17 December 2020	39	31	50	238	280
20 November–3 December 2020	31	26	43	160	245
6 November–19 November 2020	37	28	268	144	101
23 October–5 November 2020	14	3	234	137	51
9 October–22 October 2020	1	8	115	402	1
25 September–8 October 2020	0	2	0	8	0
11 September–24 September 2020	0	0	0	3	1
28 August–10 September 2020	0	0	0	2	0
14 August–27 August 2020	0	1	0	6	4
31 July–13 August 2020	2	0	0	0	0
17 July–30 July 2020	0	0	0	0	0
3 July–16 July 2020	0	0	0	0	0
19 June–2 July 2020	0	0	0	0	0
5 June–18 June 2020	0	0	0	0	0
22 May–4 June 2020	0	0	0	0	0
8 May–21 May 2020	0	0	0	0	0
25 January–7 May 2020 *	40	25	11	32	13
Total	206	143	786	1377	948

* Note: Does not adhere to the 14-day period reporting timeframe.

**Table 2 ijerph-18-06566-t002:** COVID-19 cases from 25 January–21 December 2020 for 10 districts in the Perak State, Malaysia.

14-Days Period	Districts in Perak									
	Batang Padang	Manjung	Kinta	Kerian	Kuala Kangsar	Larut dan Matang	Hilir Perak	Ulu Perak	Perak Tengah	Kampar
	Year 2010 Population Density (Person per sq km): 64	Year 2010 Population Density (Person per sq km): 204	Year 2010 Population Density (Person per sq km): 574	Year 2010 Population Density (Person per sq km): 192	Year 2010 Population Density (Person per sq km): 61	Year 2010 Population Density (Person per sq km): 155	Year 2010 Population Density (Person per sq km): 116	Year 2010 Population Density (Person per sq km): 14	Year 2010 Population Density (Person per sq km): 78	Year 2010 Population Density (person per sq km): 144
18 December–31 December 2020	18	2	129	13	1	59	189	0	16	0
4 December–17 December 2020	17	2	122	2	5	6	656	0	2	1
20 November–3 December 2020	3	1	534	3	15	9	235	4	1	39
6 November–19 November 2020	1	2	113	4	129	121	27	0	0	1
23 October–5 November 2020	10	2	20	13	2	17	0	2	0	0
9 October–22 October 2020	7	16	9	24	0	76	4	4	2	0
25 September–8 October 2020	0	0	3	0	0	9	0	0	1	0
11 September–24 September 2020	0	0	0	0	0	0	0	0	0	0
28 August–10 September 2020	0	0	0	0	0	0	0	0	0	0
14 August–27 August 2020	0	0	0	0	0	0	0	0	0	0
31 July–13 August 2020	0	0	2	3	0	0	0	0	0	0
17 July–30 July 2020	0	0	0	0	0	0	0	0	0	0
3 July–16 July 2020	0	0	0	0	0	0	0	0	0	0
19 June–2 July 2020	0	0	0	0	0	0	0	0	0	0
5 June–18 June 2020	0	0	1	0	0	0	0	0	0	0
22 May–4 June 2020	0	0	1	0	0	0	0	0	0	0
8 May–21 May 2020	0	2	0	0	0	0	0	0	0	0
25 January–7 May 2020 *	9	23	95	19	4	19	65	6	11	2
Total	65	50	1029	81	156	316	1176	16	33	43

* Note: Does not adhere to the 14-day period reporting timeframe.

**Table 3 ijerph-18-06566-t003:** COVID-19 cases from 25 January–21 December 2020 for 31 districts in the Sarawak State, Malaysia (Part 1).

14-Days Period	Districts in Sarawak									
	Kuching	Bau	Lundu	Samarahan	Serian	Simunjan	Sri Aman	Lubok Antu	Betong	Saratok
	Year 2010 Population Density (Person per sq km): 321	Year 2010 Population Density (Person per sq km): 60	Year 2010 Population Density (Person per sq km): 18	Year 2010 Population Density (Person per sq km): 210	Year 2010 Population Density (Person per sq km): 44	Year 2010 Population Density (Person per sq km): 17	Year 2010 Population Density (Person per sq km): 28	Year 2010 Population Density (Person per sq km): 9	Year 2010 Population Density (Person per sq km): 24	Year 2010 Population Density (Person per sq km): 27
18 December–31 December 2020	2	0	0	0	0	0	0	0	0	0
4 December–17 December 2020	0	0	0	0	0	0	0	0	0	0
20 November–3 December 2020	3	0	0	0	1	0	0	0	0	0
6 November–19 November 2020	97	0	0	0	0	0	0	0	0	0
23 October–5 November 2020	124	0	3	0	0	0	0	0	0	0
9 October–22 October 2020	12	0	0	0	0	0	0	1	0	0
25 September–8 October 2020	3	0	0	0	0	0	0	0	0	0
11 September–24 September 2020	1	0	0	0	0	0	0	0	0	0
28 August–10 September 2020	1	0	0	0	0	0	0	0	0	0
14 August–27 August 2020	4	0	0	0	0	0	0	0	0	0
31 July–13 August 2020	0	0	0	0	0	0	0	0	0	0
17 July–30 July 2020	51	0	1	4	0	0	0	0	0	0
3 July–16 July 2020	8	0	0	0	0	0	0	0	0	0
19 June–2 July 2020	1	0	0	0	0	0	0	0	0	0
5 June–18 June 2020	6	0	0	1	0	0	0	0	0	0
22 May–4 June 2020	4	0	0	1	1	0	0	0	0	0
8 May–21 May 2020	3	0	0	0	3	0	0	0	0	0
25 January–7 May 2020 *	325	5	1	98	17	7	3	0	16	0
Total	645	5	5	104	22	7	3	1	16	0

* Note: Does not adhere to the 14-day period reporting timeframe.

**Table 4 ijerph-18-06566-t004:** COVID-19 cases from 25 January–21 December 2020 for 31 districts in the Sarawak State, Malaysia (Part 2) (continued).

14-Days Period	Districts in Sarawak										
	Sarikei	Maradong	Daro	Julau	Sibu	Dalat	Mukah	Kanowit	Bintulu	Tatau	Kapit
	Year 2010 Population Density (Person per sq km): 57	Year 2010 Population Density (Person per sq km): 40	Year 2010 Population Density (Person per sq km): 15	Year 2010 Population Density (Person per sq km): 9	Year 2010 Population Density (Person per sq km): 108	Year 2010 Population Density (Person per sq km): 20	Year 2010 Population Density (Person per sq km): 16	Year 2010 Population Density (Person per sq km): 13	Year 2010 Population Density (Person per sq km): 25	Year 2010 Population Density (Person per sq km): 6	Year 2010 Population Density (Person per sq km): 4
18 December–31 December 2020	0	0	0	0	1	0	0	0	1	0	0
4 December–17 December 2020	0	0	0	0	1	0	0	0	0	0	0
20 November–3 December 2020	0	0	0	0	0	0	0	0	0	0	0
6 November–19 November 2020	0	0	0	0	0	0	0	0	0	0	0
23 October–5 November 2020	0	0	0	0	0	0	0	0	0	0	0
9 October–22 October 2020	0	0	0	0	1	0	0	0	4	0	0
25 September–8 October 2020	0	0	0	0	2	0	0	0	1	0	0
11 September–24 September 2020	0	0	0	0	0	0	0	0	0	0	0
28 August–10 September 2020	0	0	0	0	0	0	0	0	0	0	0
14 August–27 August 2020	0	0	0	0	0	0	0	0	2	0	0
31 July–13 August 2020	0	0	0	0	0	0	0	0	0	0	0
17 July–30 July 2020	0	0	0	0	0	0	0	0	0	0	0
3 July–16 July 2020	0	0	0	0	0	0	0	0	0	0	0
19 June–2 July 2020	0	0	0	0	0	0	0	0	3	0	0
5 June–18 June 2020	0	0	0	0	0	0	0	0	5	0	0
22 May–4 June 2020	0	0	0	0	0	0	0	0	0	0	0
8 May–21 May 2020	0	0	0	0	1	0	0	0	0	0	0
25 January–7 May 2020 *	6	0	0	0	6	0	1	0	12	0	0
Total	6	0	0	0	12	0	1	0	28	0	0

* Note: Does not adhere to the 14-day period reporting timeframe.

**Table 5 ijerph-18-06566-t005:** COVID-19 cases from 25 January–21 December 2020 for 31 districts in the Sarawak State, Malaysia (Part 3) (continued).

14-Days Period	Districts in Sarawak									
	Song	Belaga	Miri	Marudi	Limbang	Lawas	Matu	Asajaya	Pakan	Selangau
	Year 2010 Population Density (Person per sq km): 5	Year 2010 Population Density (Person per sq km): 2	Year 2010 Population Density (Person per sq km): 62	Year 2010 Population Density (Person per sq km): 3	Year 2010 Population Density (Person per sq km): 12	Year 2010 Population Density (Person per sq km): 10	Year 2010 Population Density (Person per sq km): 11	Year 2010 Population Density (Person per sq km): 103	Year 2010 Population Density (Person per sq km): 16	Year 2010 Population Density (Person per sq km): 6
18 December–31 December 2020	0	0	0	0	0	1	0	0	0	0
4 December–17 December 2020	0	0	0	0	0	0	0	0	0	0
20 November–3 December 2020	0	0	1	0	0	0	0	0	0	0
6 November–19 November 2020	0	0	5	0	0	1	0	0	0	0
23 October–5 November 2020	0	0	11	0	1	0	0	0	0	0
9 October–22 October 2020	0	0	0	0	0	0	0	0	0	0
25 September–8 October 2020	0	0	0	0	0	3	0	0	0	0
11 September–24 September 2020	0	0	0	0	0	0	0	0	0	0
28 August–10 September 2020	0	0	0	0	0	0	0	0	0	0
14 August–27 August 2020	0	0	0	0	0	0	0	0	0	0
31 July–13 August 2020	0	0	0	0	0	0	0	0	0	0
17 July–30 July 2020	0	0	0	0	0	0	0	0	0	0
3 July–16 July 2020	0	0	0	0	0	0	0	0	0	0
19 June–2 July 2020	0	0	0	0	0	0	0	0	0	0
5 June–18 June 2020	0	0	0	0	0	0	0	0	0	0
22 May–4 June 2020	0	0	0	0	0	0	0	0	0	0
8 May–21 May 2020	0	0	0	0	0	0	0	0	0	0
25 January–7 May 2020 *	0	0	23	0	9	1	1	3	0	0
Total	0	0	40	0	10	6	1	3	0	0

* Note: Does not adhere to the 14-day period reporting timeframe.

**Table 6 ijerph-18-06566-t006:** Summary of the results from both analytical models.

Correlations			
	Population Density	Cumulative Cases	Infection Rates
Population Density	1	0.644 ** (0.415)	0.216 ** (0.047)
		0.000	0.005
	143	143	143

**. Correlation is significant at the 0.01 level (1-tailed). (). R^2^.

## Data Availability

Not applicable.

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
