# Peer review of "The COVID-19 Pandemic Situation in Malaysia: Lessons Learned from the Perspective of Population Density"

_ijerph, 2021, doi:10.3390/ijerph18126566_

Round 1

Reviewer 1 Report

The authors investigated the potential link between population density and cases as well as the infection rate of COVID-19 using data collected from the national databases of Malaysia. They reported that highly dense population have higher COVID-19 cases and infection rate. Indeed, they reported that 42% of COVID-19 cases and 5% of COVID-19 infection rate (measured as the cumulative number of COVID-19 cases) could be associated with the density of the population in Malaysia. The authors performed a regression analysis and drew their conclusion on the weight of the Pearson’s coefficients (R) and coefficient of determination (R-squared). Overall, this is a very well-presented report with potential for consideration in the public health decisions on pre-emptive steps to prevent future outbreaks. However, I have suggested some changes that could improve the quality of the findings.

Major

  1. It is hard not to consider the fact that this data is collected under controlled condition i.e. lockdown measures and movement restriction. Indeed, it will be reasonable to assume that the true effect of the population density will be hard to evaluate because the preventative measures would have flattened out the relationships. Could you discuss this effect and mention the possibility of the introduced measures being a confounder of this study?
  2. Lines 122-123: Various studies have been published on the relationship between COVID-19 outcomes and co-morbidities and deserves to be cited here (https://doi.org/10.18632/aging.103000, https://doi.org/10.3390/tropicalmed5020080, https://doi.org/10.1371/journal.pone.0233147 etc).
  3. Does the correlation between population density and covid-19 cases and infection rate still stand if we correct the rates and cases for the total number of people in each state, such that the cases are expressed as per 1000 people? The authors mentioned this when they stated that “When the number of infections is taken as a percentage of the total population, a large population does not necessarily correlate with a higher rate of COVID-19 infections”. Indeed this begs the question of whether the observed correlation is truly significant and increases the probability that there may well be other factors associated with infection rate and case number of COVID-19.

Could you expatiate on this and mention this as another possible limitation and also warn readers about the interpretation and generalization of this result?

Minor

  1. The legend of fig. 1 is presented in green, yet there are no green lines on the map. Please update this legend to reflect what is shown in the figure.
  2. Abstract: can you clarify this statement “The data of these three variables covered were between 19 January 2020 and 31 December 2020. Based on the observation, districts that are dense associated with highly inter-connected geographically, socio-economically and infrastructurally tend to experienced spikes in COVID-19 cases in the weeks following the first emergence of a COVID-19 hotspot” ?
  3. Further changes have been suggested within the attached manuscript (see PDF Comments)

Author Response

We appreciate the time and effort that you have dedicated to providing your valuable feedback on our manuscript. We are grateful to your insightful comments. We have been able to incorporate changes to reflect most of the suggestions provided by the reviewer. As suggested by editor, the changes were marked up with the “Track Changes” feature for your ease of reference. However, it is advised to use the clean version of the revised paper (non-track changes) for better readability and more accurate reference in terms of section and page.

Please find below a point-by-point response to the reviewer comments and concerns.

Comment 1

It is hard not to consider the fact that this data is collected under controlled condition i.e. lockdown measures and movement restriction. Indeed, it will be reasonable to assume that the true effect of the population density will be hard to evaluate because the preventative measures would have flattened out the relationships. Could you discuss this effect and mention the possibility of the introduced measures being a confounder of this study?

Response

Agreed. It is true that a large majority of the data was collected in either a full- or semi-lockdown environment, as the Malaysian government is still employing such measures to reduce COVID-19 infection rates. Therefore, we have expanded upon our disclaimer in Section 7 (Conclusion), page 19, paragraph 1 to include the confounding effect highlighted above. To explicate further, the lockdown measures imposed during the study period (i.e., data collected from January 25, 2020 to December 31, 2020) were generally uniform throughout the nation; that means the confounding factors or effects of lockdown can be considered homogeneous, if not constant, on COVID-19 cases. Besides, apart from noting the possibility of other confounding variables, the statistically significant results (not by chance) of this study have sufficiently indicated that there is an association between population density and COVID-19 severity.

Comment 2

Lines 122-123: Various studies have been published on the relationship between COVID-19 outcomes and co-morbidities and deserves to be cited here (https://doi.org/10.18632/aging.103000, https://doi.org/10.3390/tropicalmed5020080, https://doi.org/10.1371/journal.pone.0233147 etc).

Response

Thank you for sharing this information. We have cited several of these papers in Section 6 (Discussions: Lessons Learned and Recommendations), page 17, paragraph 1.

Comment 3

Does the correlation between population density and covid-19 cases and infection rate still stand if we correct the rates and cases for the total number of people in each state, such that the cases are expressed as per 1000 people? The authors mentioned this when they stated that “When the number of infections is taken as a percentage of the total population, a large population does not necessarily correlate with a higher rate of COVID-19 infections”. Indeed this begs the question of whether the observed correlation is truly significant and increases the probability that there may well be other factors associated with infection rate and case number of COVID-19.

Could you expatiate on this and mention this as another possible limitation and also warn readers about the interpretation and generalization of this result?

Response

You have raised an interesting point here. The above-mentioned statement, as found in Section 5.2 (Population Density & COVID-19 Cases in Malaysia), page 15-16, paragraph 7-8, refers to the fact that, out of Malaysia’s 143 districts, seven districts of medium/low level of population density – such as Klang, Kota Kinabalu, and Hulu Langat – were revealed as anomalies when we charted population density against cumulative COVID-19 infection rates, as indicated in Figure 2. This implies that the findings applicable to 95% of the districts in Malaysia. However, we do agree that there are other factors that may affect the infection rates and cumulative COVID-19 cases in Malaysia, and we have acknowledged that in Section 7 (Conclusion), page 20, paragraph 1 (lines 9-14) and paragraph 2 (lines 1-9).

Comment 4

The legend of fig. 1 is presented in green, yet there are no green lines on the map. Please update this legend to reflect what is shown in the figure.

Response

Thank you for pointing this out. We have revised the legend of Figure 1 accordingly; all lines in the legend have been revised to brown, corresponding to the brown lines on the Malaysian map.

Comment 5

Abstract: can you clarify this statement “The data of these three variables covered were between 19 January 2020 and 31 December 2020. Based on the observation, districts that are dense associated with highly inter-connected geographically, socio-economically and infrastructurally tend to experienced spikes in COVID-19 cases in the weeks following the first emergence of a COVID-19 hotspot” ?

Response

Thank you for this suggestion. We have clarified the statement as follows:

“The data of these three variables were collected between January 19, 2020 and December 31, 2020. Based on the observations, districts that have high population densities and are highly inter-connected with neighbouring districts, whether geographically, socio-economically, or infrastructurally, tend to experience spikes in COVID-19 cases within the weeks of each other.”

Comment 6

Further changes have been suggested within the attached manuscript (see PDF comments)

Response

Thank you for your suggestions. We have incorporated them throughout our manuscript.

Reviewer 2 Report

In the manuscript entitled “The COVID-19 pandemic situation in Malaysia: Lessons learned from the perspective of population density”, the authors describe the determination of the impact of population density on the spread and severity of COVID-19 in Malyasia. Below are the suggestions to improve the manuscript.

  1. If the authors would have considered the impact of age/sex/socio-economic background/educational background of the population, it would have been a comprehensive analysis.
  2. Introduction: More recently, new strains reported in France and Finland were reported to be undetectable by standard RT-PCR tests (Robertson, 2021).
  3. How this study would be helpful in planning prevention measures against future pandemics? This should be explained clearly.

Author Response

We appreciate the time and effort that you have dedicated to providing your valuable feedback on our manuscript. We are grateful to your insightful comments. We have been able to incorporate changes to reflect most of the suggestions provided by the reviewer. As suggested by editor, the changes were marked up with the “Track Changes” feature for your ease of reference. However, it is advised to use the clean version of the revised paper (non-track changes) for better readability and more accurate reference in terms of section and page.

Please find below a point-by-point response to the reviewer comments and concerns.

Comment 1

If the authors would have considered the impact of age/sex/socio-economic background/educational background of the population, it would have been a comprehensive analysis.

Response

Thank you for this suggestion. It would have been interesting to explore those factors. However, in the case of our study, we aimed to obtain findings that can be applied across a more generalised population at a macro level, as population demographics, a different unit of analysis, may change over time. We hope that further analyses can be conducted for the above-mentioned population demographics and socioeconomics, and we have stated the limitations in Section 7 (Conclusion), page 19, paragraph 1 (lines 9-12). Besides, we believe that the current findings particularly on the effects of population is sufficiently novel for the case of Malaysia since by far there is no statistical evidence to prove the correlation.

Comment 2

Introduction: More recently, new strains reported in France and Finland were reported to be undetectable by standard RT-PCR tests (Robertson, 2021).

Response

Thank you for pointing this out. We have amended the citation and used the citation style preferred by MDPI, both for in-text citations and References.

Comment 3

How this study would be helpful in planning prevention measures against future pandemics? This should be explained clearly.

Response

You have raised an interesting point here. The findings and recommendations of this study are intended as passive prevention measures for similar air- and surface-borne diseases in the future. We have attempted to clarify this in Section 6.1 (Self-Contained Urban Planning), page 17; Section 6.2 (Managing Crowding Risk in Dense Areas), page 17-18; and Section 7 (Conclusion), page 19. To prevent the proliferation of similar diseases in the future, urban planners should ensure that a district is not over-crowded.

Reviewer 3 Report

Dear Authors,

the manuscript requires several major and minor corrections before it could be processed further:

  • In the abstract, the word 'This' is bolded while there is no need for that. Please correct it.
  • I suppose that phrase 

'In the initial phases, people with co-morbidities appeared more susceptible to COVID-19, but medical practitioners have since ascertained that seemingly healthy individuals can be asymptomatic carriers of the SARS-CoV-2 virus that causes COVID-19' is quite unnecessary here. It is commonly known that asymptomatic individuals might be the carriers and this knowledge was introduced nearly at the beginning of the pandemic. I would delete these sentences or paraphrase them

  • References and citation style are not according to the guidelines for authors and should be corrected throughout the whole text. Please correct it so that they are suitable for the journal. Besides I suppose that you were tried to use a template however the logo of the journal is not included. Please correct it and use a template properly.

  • ' Symptomatic patients are more contagious regardless of the presence of close contacts (Chen et al., 2020)' - I cannot find this citation in the references, there is no Chen et al. Please verify it and correct it. 
  • You are mentioning several vaccines however, you omit the others such as Moderna or Sputnik. If you want to just provide the results you should use 'OR' instead of 'AND' and the end of the sentence. If you want to enumarate all of the currently available vaccines, please include all of them. Besides you mention that the mutations of viruses continually appear while there are reports claiming that those vaccines are also effective against mutations of the virus thus I suppose that this sentence is quite inappropriate and misleading to the readers.
  •  In the abstract you are claiming that there is at least one report claiming that the person became infected even though he/she was vaccinated previously. There are numerous cases like this and this should be mentioned, thus this sentence should be paraphrased.
  • The introduction itself is too long and in some cases,  it is not related to the main topic of this paper – such information of course can be included in the text but rather in the discussion section instead of introduction. Moreover, it is quite chaotic and might be incomprehensible for the readers – some information is mixed with others instead of being mentioned in a correct order. Please correct it.
  • You are mentioning about the possible routes of SARS-CoV2 transmission while there are many more – see Baj et al. 2020 à https://doi.org/10.3390/jcm9061753
  • Figure 1 – the text on the figure is somewhere beyond the picture itself – see the upper border. Please correct it.
  • It would be beneficial to add information about the number of infected individuals as well as number of deaths in Malaysia during each of the waves of SARS-CoV-2 infection.
  • Paragraph no. 1.2. seems like a discussion – I suppose that some information from this subparagraph should be included in the discussion while only major and most relevant data (related to the topic of this work specifically) should be included in this section.
  • Paragraph 2.1. – you are not mentioning what sources of information you used specifically – government official sites? This should be included here because it is data collection specifically.
  • Error! Reference source not found.’ – it was found in the text in several senteces…, what does it mean?
  • Figure 2 and 3 and table 2 – text is written in italics while the text in other figures is not. Please unify it according to the guidelines for authors.
  • English should be corrected in the whole manuscript since numerous errors have been detected and require correction.

Author Response

We appreciate the time and effort that you have dedicated to providing your valuable feedback on our manuscript. We are grateful to your insightful comments. We have been able to incorporate changes to reflect most of the suggestions provided by the reviewer. As suggested by editor, the changes were marked up with the “Track Changes” feature for your ease of reference. However, it is advised to use the clean version of the revised paper (non-track changes) for better readability and more accurate reference in terms of section and page.

Please find below a point-by-point response to the reviewer comments and concerns.

Comment 1

In the abstract, the word 'This' is bolded while there is no need for that. Please correct it.

Response

Thank you for pointing this out. We have amended the styling in the abstract accordingly.

Comment 2

I suppose that phrase 

'In the initial phases, people with co-morbidities appeared more susceptible to COVID-19, but medical practitioners have since ascertained that seemingly healthy individuals can be asymptomatic carriers of the SARS-CoV-2 virus that causes COVID-19' is quite unnecessary here. It is commonly known that asymptomatic individuals might be the carriers and this knowledge was introduced nearly at the beginning of the pandemic. I would delete these sentences or paraphrase them.

Response

Agreed. We have omitted the sentence from the Introduction accordingly.

Comment 3

References and citation style are not according to the guidelines for authors and should be corrected throughout the whole text. Please correct it so that they are suitable for the journal.

Response

Thank you for pointing this out. We have amended the References and citation styles using the style file provided for Zotero on MDPI’s Instruction for Authors webpage.

Comment 4

Besides I suppose that you were tried to use a template however the logo of the journal is not included. Please correct it and use a template properly.

Response

Thank you for pointing this out. We have revised our manuscript using the proper template provided by MDPI.

Comment 5

' Symptomatic patients are more contagious regardless of the presence of close contacts (Chen et al., 2020)' - I cannot find this citation in the references, there is no Chen et al. Please verify it and correct it. 

Response

Thank you for pointing this out. The above-mentioned citation has been revised to Section 6 (Discussions: Lessons Learned and Recommendations), page 17, paragraph 1 (line 1-2), and the references have been updated as follows:

Chen, P.Z.; Bobrovitz, N.; Premji, Z.; Koopmans, M.; Fisman, D.N.; Gu, F.X. Heterogeneity in Transmissibility and Shedding SARS-CoV-2 via Droplets and Aerosols; Infectious Diseases (except HIV/AIDS), 2020;

Comment 6

You are mentioning several vaccines however, you omit the others such as Moderna or Sputnik. If you want to just provide the results you should use 'OR' instead of 'AND' and the end of the sentence. If you want to enumarate all of the currently available vaccines, please include all of them. Besides you mention that the mutations of viruses continually appear while there are reports claiming that those vaccines are also effective against mutations of the virus thus I suppose that this sentence is quite inappropriate and misleading to the readers.

Response

Agreed. We have revised our statement by using ‘or’ instead of ‘and’. In view of the latest information regarding the virus, we have revised our statements accordingly in Section 1 (Introduction), page 1, paragraph 2 (lines 1-3).

Comment 7

In the abstract you are claiming that there is at least one report claiming that the person became infected even though he/she was vaccinated previously. There are numerous cases like this and this should be mentioned, thus this sentence should be paraphrased.

Response

Thank you for your suggestion. We have revised our manuscript to state that existing vaccines only reduce the risk and severity of COVID-19 infection, so that it no longer implies that the vaccine confers immunity to the virus. The sentence has been amended in Section 1 (Introduction), page 1, paragraph 2 (line 6-7).

Comment 8

The introduction itself is too long and in some cases, it is not related to the main topic of this paper – such information of course can be included in the text but rather in the discussion section instead of introduction. Moreover, it is quite chaotic and might be incomprehensible for the readers – some information is mixed with others instead of being mentioned in a correct order. Please correct it.

Response

Thank you for pointing this out. We agree with your comment. Therefore, we have taken the following steps to clarify and streamline the introduction:

·         Broke up the Introduction into more Sections (Sections 2 and 3 instead of Sections 1.1 and 1.2);

·         Paraphrased or omitted sentences, as suggested in earlier comments;

·         Shifted some of the information from the Introduction to the Discussion section, as recommended (see Section 6 (Discussions: Lessons Learned and Recommendations).

Comment 9

You are mentioning about the possible routes of SARS-CoV2 transmission while there are many more – see Baj et al. 2020 à https://doi.org/10.3390/jcm9061753

Response

Agreed. We have incorporated the information in our manuscript in Section 1 (Introduction), page 1, paragraph 3.

Comment 10

Figure 1 – the text on the figure is somewhere beyond the picture itself – see the upper border. Please correct it.

Response

Thank you for pointing this out. We have revised the legend of Figure 1 accordingly so that it is within the image.

Comment 11

It would be beneficial to add information about the number of infected individuals as well as number of deaths in Malaysia during each of the waves of SARS-CoV-2 infection.

Response

Thank you for your suggestion. We have included the requested information on Section 2 (COVID-19 Status in Malaysia), page 2, paragraph 2 (line 14 - 20).

Comment 12

Paragraph no. 1.2. seems like a discussion – I suppose that some information from this subparagraph should be included in the discussion while only major and most relevant data (related to the topic of this work specifically) should be included in this section.

Response

Thank you for your suggestion. We have moved the information to Section 6 (Discussions: Lessons Learned and Recommendations), page 17, paragraph 1.

Comment 13

Paragraph 2.1. – you are not mentioning what sources of information you used specifically – government official sites? This should be included here because it is data collection specifically.

Response

Thank you for pointing this out. As per your suggestion, we have included our source of information for COVID-19 data, namely the official Ministry of Health Malaysia’s website, in Section 4.1 (Data Collection), page 5.

Comment 14

‘Error! Reference source not found.’ – it was found in the text in several sentences…, what does it mean?

Response

Thank you for pointing this out. It means that several cross-references in the document were lost in the process of formatting the manuscript. We have rectified this in our revised manuscript.

Comment 15

Figure 2 and 3 and table 2 – text is written in italics while the text in other figures is not. Please unify it according to the guidelines for authors.

Response

Thank you for pointing this out. We have standardised the captions styling in accordance with the IJERPH template provided on the website.

Round 2

Reviewer 1 Report

I am satiisfied with the changes made to the manuscript and recommend publication in the current format. 

Author Response

I am satisfied with the changes made to the manuscript and recommend publication in the current format. 

Reply: Thank you for the comment. Appreciate it.

Reviewer 3 Report

Dear Authors,

Thank you for correcting your manuscript according to my suggestions.

Could we ask you to add keywords since this part is missing and should be included just below the Abstract - see Guidelines for Authors. Please add is since it is very important and necessary.

Except that I have no further comments and suggestions. The manuscript has been corrected and in this form it seems that it is significantly improved.

Best wishes with your further research

A Reviewer

Author Response

Reviewer 3: Thank you for correcting your manuscript according to my suggestions.

Could we ask you to add keywords since this part is missing and should be included just below the Abstract - see Guidelines for Authors. Please add is since it is very important and necessary.

Except that I have no further comments and suggestions. The manuscript has been corrected and in this form it seems that it is significantly improved.

Best wishes with your further research.

Reply: Thank you very much for the suggestions. We have added all relevant keywords into the revised manuscript.